# Complementary Feeding: Pitfalls for Health Outcomes

**DOI:** 10.3390/ijerph17217931

**Published:** 2020-10-29

**Authors:** Enza D’Auria, Barbara Borsani, Erica Pendezza, Alessandra Bosetti, Laura Paradiso, Gian Vincenzo Zuccotti, Elvira Verduci

**Affiliations:** Department of Pediatrics, Vittore Buzzi Children’s Hospital, University of Milan, 20122 Milan, Italy; barbara.borsani@unimi.it (B.B.); erica.pendezza@unimi.it (E.P.); alessandra.bosetti@asst-fbf-sacco.it (A.B.); laura.paradiso@unimi.it (L.P.); gianvincenzo.zuccotti@unimi.it (G.V.Z.); elvira.verduci@unimi.it (E.V.)

**Keywords:** complementary feeding, infant nutrition, prevention, health outcomes, healthy growth, dietary habits, obesity

## Abstract

The term complementary feeding is defined as the period in which a progressive reduction of breastfeeding or infant-formula feeding takes place, while the infant is gradually introduced to solid foods. It is a crucial time in the infant’s life, not only because of the rapid changes in nutritional requirements and the consequent impact on infant growth and development, but also for a generation of lifelong flavor preferences and dietary habits that will influence mid and long-term health. There is an increasing body of evidence addressing the pivotal role of nutrition, especially during the early stages of life, and its link to the onset of chronic non-communicable diseases, such as obesity, hypertension, diabetes, and allergic diseases. It is clear that the way in which a child is introduced to complementary foods may have effects on the individual’s entire life. The aim of this review is to discuss the effects of complementary feeding timing, composition, and mode on mid and long-term health outcomes, in the light of the current evidence. Furthermore, we suggest practical tips for a healthy approach to complementary feeding, aiming at a healthy future, and highlight gaps to be filled.

## 1. Background

An increasing amount of scientific evidence suggests that during the first thousand days of life, nutrition, lifestyle behaviors, and other environmental factors exert an important role on physiology, function, health, and performance in the subsequent phases of life [1].

What happens during the first thousand days of life may influence long-term health, determining the onset risk of excess weight, obesity, and other non-communicable diseases like hypertension, cardiovascular disease, and type 2 diabetes [2,3].

Increasing evidence shows that metabolic events occurring during limited and sensitive times of prenatal and postnatal development have important modulating effects on health in later life, which is a concept often referred to as “programming”, or “metabolic programming” [4]. One of the hypotheses behind this effect is the rapid postnatal catch-up growth, which is influenced by the early nutrition pattern adopted [5]. Thus, diet in early infancy has an important role on growth pattern and development, but it has also a key role in setting up flavor preferences and behaviors, which may in turn lead to the development of being overweight or obesity.

Consequently, the first two years of life are an important period in which to begin healthy infant feeding practices in order to promote healthy growth.

Complementary feeding (CF) is a critical window, not only for the rapid changes in nutritional requirements and the consequent impact on infant growth and development, but also for the generation of lifelong flavor preferences and dietary habits, that can influence longer-term health [6].

The term CF describes a period of time in which there is a gradual reduction of frequency and volume of breast milk or formula together with the introduction of complementary foods (CFs).

According to the WHO definition, the introduction of CFs is needed to ensure optimal energy and nutrient intake when “breast milk alone is no longer sufficient to meet the nutritional requirements in terms of energy and nutrients of infants” [7]; therefore, exclusive breastfeeding is recommended for the first six months of age and complementary breastfeeding up to two years of age.

The timing of solid foods introduction is undoubtedly one of the most readily modifiable aspects of infant nutrition and the debate on the optimal age of solid food introduction is still open.

The term “complementary feeding” or “weaning” refers to all solid and liquid foods other than breast milk or infant formula. This definition has been adopted by the European Society for Paediatric Gastroenterology, Hepatology and Nutrition (ESPGHAN) and other international societies (The UK Scientific Advisory Committee on Nutrition (SACN), the United States Department of Agriculture (USDA) and the American Academy of Pediatrics (AAP) [8,9,10,11].

On the basis of the available evidence, current guidelines recommend that CFs should not be introduced before 4 months, but they should not be delayed beyond 6 months of age [8,12,13]. The timing of CF introduction coincides with the changes in nutritional requirements and the physiological maturation of renal, gastrointestinal, and neurological systems, that occur during infant development [7,8].

Alongside the “traditional” approach to solid food introduction, alternative types of complementary feeding have been emerging in recent decades, such as plant-based or baby-led weaning (BLW). These types of weaning fall out of the scope of the present review, as both they have been addressed in recent and systematic reviews [6,14,15,16,17].

The aim of this article is to discuss the effects of CF timing, composition, and mode on mid and long-term health outcomes. We also propose practical advice on how to lay the basis for healthy CF in order to promote healthy growth.

## 2. Timing of Weaning and Pitfalls for Health Outcomes

### 2.1. Excess Weight and Obesity

The ESPGHAN position paper stated that the introduction of complementary foods from 4 to 6 months of age was not associated with increased growth or adiposity during childhood [8]. Several studies have investigated one aspect of this multifaceted issue, that is, the focus on the relationship between the early introduction of weaning foods and weight gain during childhood as well as the risk of developing overweight issues and obesity. However, to date, this correlation is still controversial.

During the last decade evidence seemed to agree that the introduction of solid foods before 4 months appeared to be associated with a greater risk of obesity [18,19].

A possible theory believes that the introduction of CFs before the fourth month of age (3 months) could reduce the rate of exclusive breastfeeding, which in turn may have a negative impact on health outcomes such as a higher risk of obesity [20]. On the other hand, from the fourth month of life onwards, the timing of introduction did not appear to have an effect on the future risk of obesity.

A systematic review of interventional and observational evidence on complementary feeding timing and infant growth concluded that introduction of solid foods between 3 and 6 months appears to have neutral effects on infant growth [21].

A more recent systematic review by English et al., considering randomized controlled trial studies, prospective cohort studies, retrospective cohort studies, and case–control studies, which were published from January 1980 to July 2016, found limited evidence that the introduction of CFBs before 4 months of age may be associated with higher odds of excess weight or obesity [22].

Regarding the introduction at age ≥7 months, authors concluded that the evidence is insufficient to determine the relationship between late introduction and growth, size, and body composition outcomes [22]. In agreement with the review by English et al., the ESPGHAN position paper, after analyzing the available evidence, also concluded that the introduction of solid foods before 4 months of life may be associated with increased later obesity risk up to preschool age.

By contrast, in 2019, the EFSA released an opposing opinion regarding early introduction of CFs (from <1 to <6 months) in relation to anthropometric outcomes [12]. Specifically, and with a high level of confidence, the Panel stated that the introduction of CFs at 3–4 months of age, compared to 6 months of age, has no effect on body weight, body length, head circumference, BMI, and body composition at any age. With a moderate level of confidence, no association was detected between the timing of CF introduction and overweight problems and obesity [12].

More recently, a new prospective longitudinal study on 1013 children turned the attention back to this topic. Authors found associations between CF introduction before 4 months of age and adiposity measurements in breastfed and formula-fed children from mid childhood through to early adolescence. Stronger associations were observed in formula-fed children than breastfed ones [23].

Given the opposite conclusions reached, no recommendations can be made regarding the correlation between early introduction of CFs and the subsequent risk of developing excess weight or obesity. Additional research is needed on this topic. In particular, future research should include randomized controlled trials with long-term follow-up in order to definitively clarify this important issue.

### 2.2. Blood Pressure

With regard to blood pressure (BP), a recent retrospective plus cross-sectional study suggests the possible protective role of CF introduction between 5 and 6 months. Children who are introduced to CFs within this period of time have a lower mean systolic and diastolic blood pressure than children with early (before 5 months) or delayed (after 6 months) introduction of CFs [24]. Therefore, early or late CF introduction might constitute a risk factor for hypertension onset.

This aspect has also been investigated by the EFSA Expert Panel, which released a meta-analysis report, including four prospective cohort studies that have shown a statistically significant association between earlier timings of CF (<3 to <6 months, compared to >6 months) and higher systolic and diastolic blood pressure from 5 to about 7 years of age.

Nevertheless, the observed mean difference between the groups with earlier and later introduction of CFs was small; therefore, it was concluded that these differences are unlikely to influence the risk of cardiovascular disease later in life [12]. Further studies are needed in order to assess the long-term effect of the type of feeding, time, and nature of the weaning diet.

### 2.3. Food Preferences and Eating Behaviors

There is evidence showing that infants who are introduced to CFs before 4 months of age are more likely to consume unhealthier food at 1 year of age, even after correction for socio demographic characteristics. Accordingly, a large prospective cohort study demonstrated that children who were introduced to solid foods before 4 months of life were less able to recognize satiety signals at 5 years of age, whereas the introduction to solid foods after 6 months of life was associated with less food enjoyment and food responsiveness [25].

These results have been partially confirmed by the recent review conducted by EFSA, which highlights how the introduction of CFs before 3 or 4 months of age, compared to 6 months of age and thereafter, is associated with some less desirable eating behaviors, such as lower satiety responsiveness, higher feeding difficulties, and lower likelihood to have a positive eating pattern [12].

### 2.4. Developmental Milestones

According to the definition of The Centres for Disease Control and Prevention (CDC) [26], developmental milestones are a set of behaviors, skills, or abilities that are demonstrated by specified ages during infancy and early childhood in typical development.

Regarding this topic, a recent systematic review [27] investigated the relationship between the timing of CFB introduction and communication, cognitive and motor development. It concluded that there is insufficient evidence to draw a conclusion about the relationship between timing of introduction of CFBs and developmental milestones.

### 2.5. Food Allergy

Current guidelines recommend not to delay the introduction of allergenic foods beyond 4–6 months of age [8,28,29,30]. While there is consensus not to delay the introduction of allergenic foods, the optimal timing of introduction is still debated.

In high-risk infants, e.g., infants with a personal history of atopic dermatitis and/or having a first-degree relative with a history of allergic diseases, the introduction of peanuts from 4 to 11 months of age may be beneficial to prevent peanut allergy [31]. With regard to other allergenic foods, such as milk or fish, data from interventional studies are still limited and do not indicate an association between timing of introduction and allergy or sensitization onset [32].

Apart from the introduction of single foods, it seems that an increased dietary diversity may have a role in preventing food allergies.

The PASTURE/EFRAIM trial by Roduit et al. found that an increased diversity of complementary foods introduced within the first 6 and 12 months of age (considering cereals, bread, meat, fruits/vegetables, cake, and yogurt) was associated with a reduced risk of food allergy onset [33]. Similarly, a recent trial by Venter et al. demonstrated that infants with a dietary pattern which included more food groups at 6 and 9 months of age had a lower risk of developing food allergies during childhood. In particular, authors found that the introduction of each additional food allergen by the age of 12 months reduced by one third the odds of developing a food allergy during the first decade of life [34].

The mechanism through which dietary diversity acts on the development of food allergies is still unclear; higher food variety may favor the intake of nutrients such as dietary fiber or omega-3 fatty acids, which may in turn be related to allergic disease prevention, or which may have a role in increasing gut microbiome diversity. In addition, the consumption of a wider range of foods increases exposure to food allergens, possibly favoring tolerance development [35].

## 3. Macronutrients Composition of CF and Health Outcomes: How Much Is Not Too Much?

The intake of macronutrients and micronutrients during the CF period should be optimal so as to ensure appropriate infant growth and to promote health outcomes in early and later life.

Energy is the main determinant for fat deposition and daily intake should be correctly distributed among macronutrients without exceeding nutritional needs [36].

### 3.1. Macronutrients: Proteins

A large amount of evidence has recognized how rapid or excessive weight gain (RWG) during early sensitive periods of life is associated with increased adjusted odds for obesity in later childhood, adolescence and adulthood [37,38]. It has also been recognized that infant feeding habits have the potential to influence weight gain velocity and later on, obesity.

Growth patterns of breastfed infants differ from those of formula-fed infants: studies report that breastfed infants have a progressive decrease in growth indices, while formula-fed infants show a continuous increase during the first 12 months of life [39,40]. Moreover, formula-fed infants seem to gain weight more rapidly in proportion to length than breastfed ones, resulting in a progressive increase in BMI over time [41].

Breastfeeding seems to exert a protective effect against obesity development in later life, while formula feeding seems to favor high weight gain [42].

These different weight patterns may be due to several factors that are difficult to untangle, such as the different nutrition profiles of breast milk and formula milk. For example, there is a lower content of protein in human milk than in formula milk.

Breastfeeding may also promote increased self-regulation and better satiety responsiveness compared to formula feeding [43]. Moreover, breast milk contains hormones and bioactive compounds that could inhibit adipogenesis, playing a role in future obesity prevention [44]. Standard infant formulae provide comparable amounts of fat and carbohydrates, but a much higher quantity of proteins compared to human milk. Substantial evidence has shown that the higher protein content in formula milk compared to breast milk stimulates excessive growth and increases the risk of obesity [45,46]. On the other hand, a longer duration of exclusive and partial breastfeeding tends to be associated, in a dose-dependent manner, with slower growth rates in infancy [47].

On the basis of these observations, “the proteic hypothesis” argues that an excessive protein intake in infancy may induce metabolic programming of both rapid weight gain and increased obesity risk later (Early Protein Hypothesis) [37]. Specifically, high protein intake increases plasma concentrations of insulin-releasing amino acids, stimulates the secretion of insulin and insulin-like growth factors 1 (IGF-1), and enhances weight gain and body fat deposition.

A large double-blind randomized trial with long term follow-up, conducted by the European Childhood Obesity Project (CHOP Study), has demonstrated that, at 2 and 6 years of age, infants fed with standard formulae (2.9 g protein/100 kcal) had a significantly higher mean BMI than infants fed with a protein reduced formula (1.77 g protein/100 kcal). Compared to breastfed infants, children fed higher protein formula had significantly higher BMI values and BMI z-scores, whereas the growth pattern in the lower protein group was not different from the breastfed group.

Notably, the conventional infant formula feeding with high protein content induced a significantly higher fat mass index (FMI) compared to the protein reduced intervention group. The FMI in the lower protein intervention group was similar to breastfed group [46,48].

Another large cohort study of 2154 children (the Gemini Study), investigated whether a higher proportion of protein intake from energy during weaning was associated with greater weight gain, higher body mass index (BMI), and the risk of overweight issues or obesity in children up to 5 years of age. Results show that total energy from protein is associated with higher BMI and weight, but not with length between 21 months and 5 years. Interestingly, the replacement of a caloric quota provided by protein with the same caloric quota provided by carbohydrates and fats resulted in a reduction in BMI and weight [49].

In another cohort study from UK (The Avon Longitudinal Study of Parents and Children ALSPAC), dietary intake was recorded at 8 months, and infants were divided into four groups according to amount and type of milk consumption (cow’s or formula milk). Investigators found that high volumes of cow’s milk (>600 mL) in formula-fed infants determined higher BMI up to the last examination at 10 years [50].

Cow’s milk is a poor iron source and when consumed in large amounts provides an excess of protein, fat, and energy. As a high intake of cow’s milk during the CF period can increase the risk of iron deficiency anemia as well as the onset of obesity, the consumption of cow’s milk is generally not recommended during the first year of life [8].

It must be considered that during the CF period the nutritional protein requirements are different between breastfed and formula-fed infants. Moreover, the introduction of solid foods increases children’s protein intake whereas protein requirements decrease [51].

For these reasons, during the CF period, the amount of protein and the proportion of energy supplied can exceed the physiologic requirements and the recommended energy needs. For instance, in the Gemini Study [49] children were found to be consuming on average 3–5 times more than the amount of protein intake recommended by WHO.

These data, as well as those of systematic reviews [52,53], support the nutrition statement from the European Society for Paediatric Gastroenterology, Hepatology and Nutrition Committee that higher protein intake in infancy and early childhood, particularly when the energy percentage from protein at 12 months is between 15% and 20%, is associated with increased growth. For all these reasons, the upper limit at 12 months of the mean protein intake is established at 15% of total energy [8].

Regarding a possible association between protein intake and later risk of hypertension and cardiovascular disease, there is a dearth of data. A recent population-based prospective cohort study [54] showed that elevated adherence to a “health-conscious”dietary pattern at 1 year of age was associated with a lower combined cardio-metabolic risk factor score at 6 years of age.

### 3.2. Carbohydrates

The recommended mean daily intake of carbohydrates is between 45% and 60% of total energy [36]. The largest share of energy should be provided by complex carbohydrates, with a preference for starchy alimentary sources and if possible, with a low glycemic index. Since an overconsumption of energy-dense CFs may induce excessive weight gain, the intake of soluble sugar should be limited [36]. Evidence has shown that the consumption of sugar-sweetened beverages (SSB) is associated during infancy with higher odds of obesity at 6 years, compared to non-SSB consumers [55,56]. Consequently, to prevent higher energy concentration, it is important to recommend limiting dietary sugar intake with beverages and foods in infancy and early childhood. This recommendation is also reflected in a recent systematic review which highlighted how some caregivers’ practices, such as adding cereals to a bottle as well as incorrectly mixing a powder formula (e.g., more concentrated) may contribute to rapid weight gain [38].

The recommended daily intake of dietary fibers from 6 to 12 months of age has not yet been defined [57]; a moderate and constant intake does not adversely affect the energy intake [36,58].

### 3.3. Lipids

The recommended mean daily intake of fats is 40% of total energy, and should not be lower than 25% [59]. Long-chain polyunsaturated fatty acid (PUFAs) intake should be 100 mg/day, with linoleic acid (LA) accounting for4% of the daily energy intake, and alpha-linolenic acid (ALA) for 0.5% of the daily energy intake. Saturated fatty acids should be limited to as low as possible [59].

No association has been found between a high fat intake during weaning and obesity at later ages; on the contrary, a possible contributing factor of early adiposity rebound has been identified in a hyperproteic and hypolipidic diet [60]. A 2015 Cochrane review assessed the effects of fat intake in infancy on childhood body weight. The authors concluded that there was no clear association between lower total fat intake and later BMI and/or body composition. In summary, to date there is no conclusive evidence of a relationship between fat intake in the first years of life and childhood BMI or early adulthood adiposity [61].

With regard to the role of PUFAs during the complementary feeding period and their long-term effects on cardiovascular health, current evidence is weak; no association has been found between high PUFA intake during this period of infant growth and obesity and body fat content in a subsequent span of life [62].

### 3.4. Micronutrients

One of the most critical micronutrients during the CF period is iron, as its high level requirement is often difficult to ensure. As a consequence, the risk of developing iron deficiency or even iron deficiency anemia in infancy should be considered. Iron is also critical for neurodevelopment: iron deficiency is a risk factor for both short-term and long-term cognitive impairment. Insufficient dietary administration during infancy is associated with poor mental and motor development and, in addition, with poor cognition and school achievement during later childhood [63].

According to the recommendations of scientific societies, average daily iron intake varies from 6 to 11 mg/day [64]. Particular attention should be paid to exclusively breastfed infants, because breast milk cannot meet iron requirements at any age. From 6 months of age, most of the iron requirements should be met through complementary foods; for this reason, it is advisable to introduce iron-rich foods (e.g., red meat) and iron-fortified infant cereals [65]. Several studies support the timely introduction of iron-rich CFs in breastfed infants, identifying meat as a good first complementary food for all breastfed infants [65,66,67].

A recent systematic review confirms, with strong evidence, that consuming CFBs that contain substantial amounts of iron (such as meat or iron-fortified cereals) helps to maintain adequate iron status or to prevent iron deficiency during the first year of life. This is particularly important among infants with insufficient iron stores or breastfed infants who are not receiving adequate iron from another source. Instead, the benefit of these types of CFBs for infants with sufficient iron stores, such as those consuming iron-fortified infant formula, is less evident [68].

The recommended daily zinc intake is 2.9 mg/day [64]. Zinc acts as a cofactor for many enzymes; it is essential for proper growth, development, and cognitive function. It also has a role in the immune response. In infants its deficiency can lead to chronic diarrhea as well as learning and memory impairment; as the child grows, hair loss, skin diseases, growth retardation, and increased susceptibility to infections may also occur [69,70].

The introduction of meat as an early complementary food in exclusively breastfed infants seems to be associated with improved zinc status and potential benefits on psychomotor development [71].

Limited evidence suggests that consuming CFBs that contain substantial amounts of zinc (such as meat or cereals fortified with zinc) supports zinc status during the first year of life, particularly among breastfed infants who are not receiving adequate zinc from another source [68].

The recommended daily calcium and vitamin D intakes are 260 mg and 400 IU, respectively [36]. With regard to vitamin D, there is not enough evidence to determine the relationship between types and amounts of CFBs and the status of this micronutrient [68]. Good sources of calcium are milk and its derivatives, vegetables, cereals, meat, and fish. An insufficient intake of calcium and vitamin D can cause reduced bone mineral density and a higher risk for bone fractures, rickets, and osteomalacia. Moreover, calcium and vitamin D status may be linked to some chronic diseases, including obesity, type 2 diabetes mellitus, and dyslipidemia. Indeed, vitamin D also seems to have an anti-inflammatory role. On the other hand, calcium may increase body fat oxidation and fecal fat excretion, as well as reduce intestinal fat absorption, contributing to a negative energy balance [72].

Vitamin A is the name of a group of fat-soluble retinoids that can be found in foods of animal origin and in many types of fruits and vegetables containing carotenoids which are vitamin A precursors. Maintaining muscular integrity and vision is essential for growth, both general and cellular, differentiation and signaling, normal metabolism, and for immunological purposes [73].

Vitamin A deficiency (VAD) can cause impaired tissue function, which may result as critical, especially during the developmental periods of infancy and childhood [74]. In developing countries, an association between VAD and higher odds of stunted growth and growth failure, independent of potential confounders, has been highlighted [75]. In addition, supplementation with vitamin A and proteins has resulted in an increased mean height for age and weight for age z-scores, suggesting an effective role in preventing growth failure [76]. Vitamin A deficiency (VAD) can also lead to anemia and a reduced resistance to infection, which in turn can increase the risk of death from severe infections [74].

Vitamin A appears to play a central role in formation and maintenance of vision. From as early as embryonic development, vitamin A—in the form of retinal—helps set up the framework for healthy eyes [77]. VAD is the leading cause of preventable childhood blindness issues, such as dry eye (xerophthalmia), age-related macular degeneration (AMD), and impaired night vision. This is why vitamin A is considered essential for healthy vision [78]. The recommended dietary intake in the pediatric population is from 200 to 500 μg/d [64,73,79]. Vitamin A supplementation is recommended in children between the ages of 6 months and 5 years in all developing countries, while in developed countries supplementation is not indicated [80,81].

The recommended daily sodium intake is 0.4g. [36]. Infants with “lower sodium regimen” have shown a significantly lower systolic and diastolic blood pressure, both at 6 months of age and 15 years later, when compared with controls fed with “normal sodium regimen” [82].

Unlike the unclear evidence about protein intake and BP, the evidence of an association between sodium intake and BP in early life is limited but consistent [83]. Consequently, to ensure adequate development and to reduce the risk of subsequent obesity, it is essential to establish a correct and balanced diet in macro and micronutrients, mindful of the caloric and protein intake with appropriated portion sizes. No sugar or salt should be added to CFs and fruit juices or sugar sweetened beverages should be avoided [8,84].

The introduction of cow’s milk is recommended from 1 year of age onwards, although small volumes may be added to CFs [8].

Practical tips for an adequate and healthy CF are summarized in Table 1.

The suggested composition of the infant’s first meal is illustrated in Figure 1 and appropriate portions of several foods are reported in Table 2.

Recommended daily intake for macro and micronutrients are reported in Appendix A.

## 4. Mode of Feeding and Health Outcomes

### 4.1. Diet Diversity and Health Outcome

Diet diversity is defined as the number of foods or food groups consumed over a short period of time (generally 24 h). It has been identified by the WHO as one of the core indicators to assess adequacy of complementary feeding practices in children aged 6–23 months [85].

Since, during the complementary feeding period, breast milk no longer satisfies the infant’s nutritional requirements, a greater diet diversity (meaning a wide variety of foods introduced in the diet) facilitates the introduction of a broader range of potential CFs [86]. In addition, the exposure to various textures and tastes seems to support the establishment of healthy food preferences, persisting into later ages [87,88].

Diet diversity has been little examined in relation to long-term health outcomes.

Arimond and Ruel in 2004 collected data from 11 demographic surveys in African, Asian, and American infants and found that greater diet diversity (evaluated as a 7-point score given by the consumption of foods divided into seven major food groups) was associated with better nutritional status. This finding was observed despite the fact that the studies lacked uniformity and the populations had very different nutritional patterns, suggesting the strength of the association [89].

According to some studies, breastfeeding might be related to enhanced diet variety, because early exposure to a variety of flavors through breast milk may encourage the acceptance of different tastes introduced with solid foods [90,91]. On the contrary, the excessive consumption of infant formula after 12 months of life might delay or reduce the introduction of complementary foods, thus limiting the variety of foods consumed by the child [92,93].

Food items can be classified into groups that share particular characteristics. For instance, animal-derived foods such as meat and fish are rich in protein, iron, and zinc, whereas vegetables have a high content of dietary fiber and vitamins and are generally low energy-dense [94]. Fish consumption allows the child to consume a good amount of PUFAs, avoiding the risk of negative effects on cognitive development [95]. Several studies suggest that the intake of oily fish DHA or precursor fatty acids during the CF period may influence DHA status with effects on short-term visual function [8,96]. Additionally, iron and Vitamin B12, s found in animal-derived foods, are essential to support proper neurodevelopment. To ensure an adequate intake, iron-fortified infant cereals and commercial meat-based infant foods are recommended.

Since each food group provides specific micronutrients, rather than the recommendation to introduce a specific food or nutrient into the diet, the importance of consuming several tastes and textures should be emphasized. Moreover, there is increasing evidence to suggest that infants who are fed with a more varied diet have decreased body fat mass compared to infants who follow a less varied diet [97].

### 4.2. Responsive Feeding and Health Outcomes: Weight Gain and Obesity

Another important aspect to consider regarding health outcomes is the effect of the way in which infants are fed. It is also important to look at the interaction between children and parents during CF.

Responsive feeding (RF) is the most effective approach to promote a healthy growth pattern [98,99]. This method relies on the parents’ ability to understand babies’ signals that indicate, for instance, hunger or satiety. Infants have inherently good satiety signals that should not be overridden; however, parents are often more responsive to hunger signals than to satiety signals [100]. Evidence suggests that, if caregivers regularly feed infants in the absence of hunger and/or beyond their satiety, the infant’s ability to eat in response to hunger and fullness cues may be undermined [101,102,103]. Consequently, a discordant chronic feeding response, with an inability to understand or misinterpretation of children’s signals, easily leads to overfeeding and may contribute to weight gain and obesity, as well as to aggravate further problematic feeding behaviors [104].

Nonresponsive feeding practices include situations in which caregivers exert excessive pressure and authoritarian control (e.g., restriction or pressure to eat), or use food as a reward or as an emotional control tool (e.g., to calm, distract, or comfort the child). In addition, contexts in which the child can completely control the feeding situation (e.g., indulgent feeding), or the caregiver is completely uninvolved during meals (e.g., uninvolved feeding) are considered to be nonresponsive feeding practices [99]. Otherwise, a parent who allows the infant to determine timing, amount, and pacing of a meal helps the infant to develop self-regulation and secure attachment [104].

This mechanism is established early in infancy; actually, breastfed infants have more control over their milk intake, whereas formula-fed infants are often encouraged to finish a bottle. Moreover, mothers who breastfeed are more likely to adopt a RF approach with the repetition of positive behaviors during CF [43,105].

The SLIMETIME intervention [106], a randomized controlled trial focused on RF rather than diet, detected a significantly slower rate of weight gain in infants who had received a multicomponent behavioral intervention (RF and indications for introduction of solids), compared to the other groups (the group receiving only one intervention and the control group receiving no intervention), suggesting that educating parents about RF may be more beneficial than dietary advice alone. At 1 year of age infants in the intervention group had lower weight-for-length percentiles (*p* = 0.009).

The NOURISH study, a randomized long-term controlled trial, evaluated an obesity prevention intervention, which targeted feeding practices in the first year of life. This study showed that the intervention group reported less frequent use of nonresponsive feeding practices, more appropriate responses to food refusal and higher satiety responsiveness as well as lower emotional overeating, fussiness, and food responsiveness. No statistically significant group effect was found for anthropometric outcomes (BMI z-score: *p* = 0.06) or for overweight/obesity prevalence (control 16% vs. intervention 17%, *p* = 0.060). However, the authors concluded that this difference translated to a population level would represent a meaningful public health effect [107,108,109].

Furthermore, data from the INSIGHT Responsive Parenting study, a randomized controlled trial, showed that children of mothers who received an RP intervention were less likely to have dietary patterns with low intake of appropriate CFs or high intake of fruit juice and energy dense foods, regardless of feeding mode. Besides, infants following the most appropriate dietary pattern during the complementary feeding period had a modest reduction in BMI z-scores at age 3 years compared with the control group. This study demonstrates that teaching RP may also promote more healthful dietary patterns among formula-fed infants [110,111,112].

Several systematic reviews [102,113,114] are in agreement, stating that interventions with the aim of improving parental feeding practices with attention to childhood cues, can lead to encouraging results in relation to behavior change, but not to effects on weight. As regards the effects on anthropometric outcomes, more research is needed to clarify the role of RF interventions in the prevention and treatment of children overweight problems and obesity.

All these findings suggest that weight outcome is probably the consequence of an accumulation of experiences; therefore, multi component interventions promoting RP behaviors and reinforcement during developmental stages of infants may potentially contribute to achieving and maintaining a healthy growth trajectory.

### 4.3. Taste and Food Preferences

Infants naturally prefer sweet, salty, umami flavors, and energy-dense foods. They tend to reject the potentially toxic flavors of bitter and sour [115]. Thus, these preferences predispose them to initially reject “healthy foods” like complex carbohydrates and green leafy vegetables which are not sweet, salty, and energy-dense [116]. These preferences are innate to biological and genetic bases [91]; even though they represent an adaptive advantage in food-scarce environments, in the modern food environment they are maladaptive. Evidence suggests that an infant’s taste experiences begin early, initially in utero and then later during breastfeeding, where flavors from the mother’s diet are tasted in the breast milk [117].

Acceptance of basic flavors during complementary feeding may be different among breastfed and formula-fed infants. Formula-fed infants are often exposed to a constant single flavor, a predominantly sweet taste. Instead, breastfed children, in addition to the sweet taste of milk, are also exposed to varying flavors and aromas, depending on the mother’s nutrition [117].

A large amount of evidence and recent review studies confirm that the repeated exposure strategy, which consists in offering the same flavor or food frequently (8–10 exposures), is the most effective way of promoting acceptance of new foods over time. Accordingly, a repeated exposure to a varied diet can increase acceptance and eventually the liking of more nutritious foods such as fruits and vegetables [118,119,120].

However, this mechanism also occurs for unhealthy food preferences; indeed, if parents and caregivers offer sweet and salty food, they can reinforce innate flavor preferences. With regard to this topic, longitudinal studies, examining the preference for sweet solutions versus water in infants, demonstrated that infants who were regularly fed with sugar water had a preference for significantly higher concentrations of sucrose solutions 2 years later, compared to those who had no such experience. This difference was found despite the fact that, at birth, all of the infants preferred sweet solutions to water [121,122].

Another prospective study demonstrated that early dietary experience was related to salt acceptance, with only infants previously exposed to starchy table foods preferring the salty solutions at 6 months (*p* = 0.007). Infants eating starchy table foods at 6 months were more likely to lick salt from the surface of foods at preschool age (*p* = 0.007) and tended to be more likely to eat plain salt (*p* = 0.08) [123].

A recent prospective cohort study investigated how feeding patterns during CF influence future diet quality; authors demonstrated that certain CF behaviors can predict the child’s diet quality approximately 2 years later. For instance, membership in the group characterized by delayed introduction of sweets and low fruit juice intake at 1 year of age predicted higher diet quality at 3 years [124]. Consequently, in order to obtain an increased acceptance and a varied diet, caregivers should discourage innate flavor preferences of their infants by offering them healthy foods, and by not adding salt or sugar or energy-dense ingredients. Exposing children to a variety of flavors and adopting the repetition technique has the advantage of promoting children’s inclination to consume and accept new foods [125,126,127].

All these strategies are functional and are intended to direct children’s preferences towards more varied and healthy flavors and less towards unhealthy foods.

## 5. Conclusions

The complementary feeding period is a sensitive time window which can influence growth, development, food preferences, and short-term as well as long term health outcomes. During this period, it is essential to pay attention to the quality of the diet, in order to prevent nutritional deficiencies or excesses, and to set correct eating habits.

Based on the physiological maturation of renal, gastrointestinal, and neurological systems and the need to meet the different nutritional requirements of infants, CFs should be introduced in a period of time ranging from 4 to 6 months of age, in developed countries. According to the WHO recommendation, in low-income countries, exclusively breastfeeding a child may be prolonged beyond the first year of life, due to the significant advantages it has against infectious diseases.

The nutritional needs of the child should always be respected; this issue may be easier to achieve by promoting diet diversity as early as the first year of life.

In order to provide the child with a well-balanced diet, it is advisable to meet his/her caloric needs, without exceeding protein intake. Parental behaviors play a key role in establishing correct eating habits according to different ethnicity. Caregivers should always respect infant cues in order to set a functional relationship and acceptance of new foods, including bitter vegetables, and different consistencies. The repeated exposure strategy has proven effective in increasing the acceptability of foods.

Regarding allergenic foods, there is no evidence that delaying their introduction might be effective in preventing food allergy onset.

The best timing of solids food introduction is still an open question, especially regarding the effects on mid and long-term health outcomes. These should be further investigated.

This review proposes some practical advices on the basis of the best evidence. Additional research is needed to address some knowledge gaps.

Table 3 shows the areas on which future research should focus.

## Figures and Tables

**Figure 1 ijerph-17-07931-f001:**
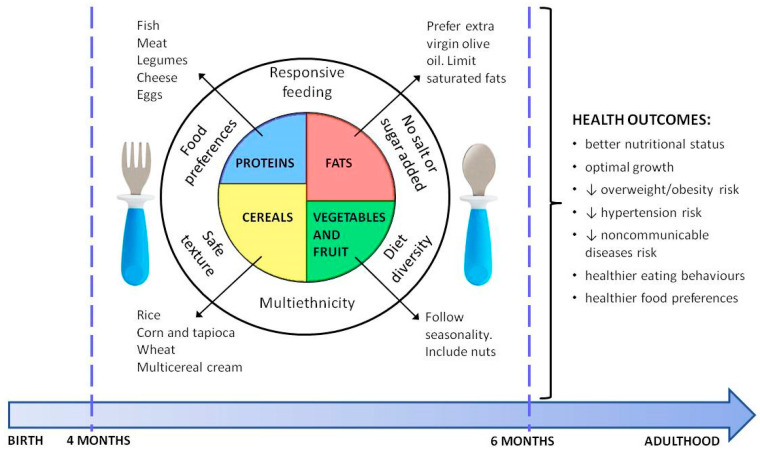
Suggested composition of the infant’s first meal and health outcomes.

**Table 1 ijerph-17-07931-t001:** Practical tips for healthy complementary feeding: “aiming at a healthy future”.

Practical Tips for Complementary Feeding
Complementary foods should not be introduced before 4 months but should not be delayed beyond 6 months During complementary feeding, if possible, continue breastfeeding for up to 2 years of age Respect the adequate daily caloric intake (70–75 kcal/kg/day) Do not introduce excessive quantities of hyper-caloric and high-protein foods, choosing instead low-energy density foods (e.g., fruits and vegetables) Propose age-appropriated portion sizes (pay attention to the protein intake; see Table 2) Do not introduce cow’s milk before 12 months of age Avoid adding salt and/or sugar to the infant’s food Avoid proposing fruit juices and sugar sweetened beverages Promote dietary diversity to propose a range of tastes and textures Foods should always be safe: ensure they have appropriate texture and consistency for the infant’s developmental level Recognize infant cues, avoid feeding to comfort and reward Try repeated exposure to foods (8–10 exposures)

**Table 2 ijerph-17-07931-t002:** Examples of portion sizes at 6 and 12 months during the complementary feeding phase.

Examples of Portion Sizes during the Complementary Feeding Phase		
	6 Months	12 Months
Cereals	Cereal creams (rice, corn and tapioca, wheat, multi-cereal cream)20 g	Baby pasta 25 g Rice 25 g Couscous 25 g
Homogenized meat	40 g (half a jar)	80 g (a jar)
Freeze-dried meat	5 g	10 g
Fresh meat	15 g	30 g
Homogenized legumes	40 g (half a jar)	80 g (a jar)
Dry decorticated legumes	10 g	15 g
Fresh legumes	25 g	40 g
Homogenized fish	40 g (half a jar)	80 g (a jar)
Fresh fish	20 g	30 g
Fresh cheese	20 g	30 g
Egg	25 g (half an egg)	50 g (an egg)
Seasonal vegetables	20 g	30 g
Extra virgin olive oil (during the day)	10 g	20 g
Vegetable broth (no added salt)	160–180 mL	
Fresh fruit	80 g	80 g

**Table 3 ijerph-17-07931-t003:** Complementary feeding: gaps to be filled.

Research Areas Regarding the Complementary Feeding Period
Nutrient needs during the complementary feeding phase should be differentiated taking into account feeding practices (as the UK British Nutrition Foundation guidelines suggest), in order to personalize the recommendations Long term studies evaluating the effects of complementary feeding on health outcomes should be conducted A dietary fiber reference intake has not yet been proposed for infants aged <1 year old. Since dietary fiber may have a prebiotic effect, with the ability to modulate gut microbiota composition, their relevance during the first year of life should be further explored, according to the different milk feeding. In addition to the recommended intake of individual nutrients, the effects of diet diversity in the first year of life on health outcomes should be further investigated.

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
