# Peer review of "Complementary Feeding: Pitfalls for Health Outcomes"

_ijerph, 2020, doi:10.3390/ijerph17217931_

Round 1

Reviewer 1 Report

I believe the document is interesting bit at the same time it has been writing in a way that it seems as a manuscript to be reading by parents and not specialist. It is to even; it is missing versatile and attracted. It is a very plain paper.

The authors used two different ways of mentioning references (89). And after .(89). Homogenized document.

They also use capital letter without a reason to do so.

English revision is definitely necessary. Use only an American or English style.

Not a consistent used of -.

Line 84. What’s RCT?

The use of LCPUFA instead of helping it confusing and wide world use of PUFA is accepted and very well known.

Add versatility to the document (Diagrams, figures, will make it more readable, even for parents.

References:

Add Doi numbers to all references.

Remove the months and editorial from journal (they are not very helpful).

Homogenize titles. Again the use of capitals letters in inconsistent.

Check all references for typing mistakes.

Author Response

We would like to thank the reviewer for the valuable comments that allowed us to improve the quality of our paper. As the reviewer suggested, we added a figure and two more tables (Figure 1, Table 3 and Table 4), to make the paper more attractive.

We provided to describe and correct the abbreviation used in the text, as requested. The English language has been edited according to an English style throughout the text, by a native speaker, as suggested.  The paragraph about timing of weaning and overweight and obesity has been re-written, emphasizing the contradictory results of the studies, as also requested by the reviewer 2. Furthermore, the conclusions have been also partly edited.

All  typo errors  listed below have been corrected, as  following:

The authors used two different ways of mentioning references (89). And after .(89). Homogenized document. - We homogeneized  the ways of mentioning references.

They also use capital letter without a reason to do so. - We left capital letters only when necessary.

Not a consistent used of -. - We corrected them.

Line 84. What’s RCT? - We described the abbreviation (randomized controlled trial).

The use of LCPUFA instead of helping it confusing and wide world use of PUFA is accepted and very well known. - We replaced LCPUFA with PUFAs throughout the text.

References:

Add Doi numbers to all references. - We added Doi numbers, where available.

Remove the months and editorial from journal (they are not very helpful). - We removed them.

Homogenize titles. Again the use of capitals letters in inconsistent. - We homogeneized capital letters.

Reviewer 2 Report

It’s a very good revision about the subject.

Nevertheless, about the studies explained between 84 and 113, the contradictions may be more emphasised. There are contradictory results in the studies 8, 12, 20, 21,22 in the references cited and described in the paper. So, in conclusion 115-119 the authors demonstrate controversial results, but these contradictions must be emphasised.

By consequence in conclusions 492,493 there is a contradiction, which may be explained.

The making reference of 299 is yet signalled.

Author Response

Thanks to the reviewer for the positive comment on the review.

As suggested, the paragraph about timing of weaning and overweight and obesity has been completely re-written, as suggested. (lines 84 to 113, now 84 to 125). We also emphasized the contradictory results of  the studies presented in this section, especially focusing on early introduction of solid foods which is still an open question. Accordingly, the conclusions have also been edited (Lines 540 to 546).

We have included the reference of line 299 (as requested also by Reviewer 3).

Reviewer 3 Report

Complementary feeding is defined as the process starting when breast milk alone is no longer sufficient to meet the nutritional requirements of infants, and therefore other foods and liquids are needed, along with breast milk. The transition from exclusive breastfeeding to family foods – referred to as complementary feeding – typically covers the period from 6–24 months of age, even though breastfeeding may continue to two years of age and beyond. This is a critical period of growth during which nutrient deficiencies and illnesses contribute globally to higher rates of undernutrition among children under five years of age. This review about "Complementary feeding: pitfalls for health out comes" will provide ample information. This review has minor error need to be fix before publication. 

  • COMMENTS
  • It is important to include a graph that shows the gaps to be filled by complementary food for a breast fed child. This will increase the scope of this review
  • Author has not included any information regarding vitamin A, Because Vitamin A is important for normal growth and vision for children.
  • Including daily reference intake of micro nutrients, macro nutrients and lipds etc.. as table based on age will increase the scope of this review.
  • Page-1 Line 17 there is repeated words
  • Line 299 reference need to be included.

Author Response

Thanks to the reviewer for the positive comment on the review.

As the reviewer suggested, we added a paragraph about vitamin A functions (lines 344 to 364).
We also added a table comparing daily reference intakes of the main nutrients at different ages, as recommended by Scientific Societies (Table 3 in Supplemental materials).

A table summarizing  research gaps about the complementary feeding period  has been added (Table 4).

All the typo errors  listed below have been eliminated, as following:

Page-1 Line 17 there is repeated words - We deleted the repeated words.

Line 299 reference need to be included - We included the reference (as requested also by Reviewer 2).

Round 2

Reviewer 1 Report

The authors have done all previous comments to their manuscript. So I believe it is acceptable now in the present form.